# Immediate and short-term effects of neurodynamic techniques on hamstring flexibility: A systematic review with meta-analysis

Sergio Núñez de Arenas-Arroyo[1], Vicente Martínez-Vizcaíno[1,2]*, Ana Torres-Costoso[3], Sara Reina-Gutiérrez[1], Bruno Bizzozero-Peroni[1,4], Iván Cavero-Redondo[1,2]

1 Health and Social Research Center, Universidad de Castilla- La Mancha, Cuenca, Spain, 2 Facultad de Ciencias de la Salud, Universidad Autónoma de Chile, Talca, Chile, 3 Facultad de Fisioterapia y Enfermería, Universidad de Castilla-La Mancha, Toledo, Spain, 4 Instituto Superior de Educación Física, Universidad de la República, Rivera, Uruguay

* Vicente.Martinez@uclm.es

## Abstract

### Background

Good hamstring flexibility(HF) is crucial for sports performance and health, with injuries having an economic impact on healthcare and sports teams. Therefore, our objectives were to estimate the effect of neurodynamic techniques on HF and to compare the effect of these techniques with static stretching.

### Methods

We systematically searched the Cochrane, MEDLINE(via PubMed), Scopus, Web of Science and Sportdiscus databases for RCTs comparing neurodynamic interventions with control intervention or with static stretching exercises for HF in adults with limited HF. We conducted a random-effects meta-analysis with subgroup analyses according to the type of comparison group(control group or static stretching exercises) and total number of sessions. Furthermore, to reflect the variation in genuine therapy effects in different scenarios, including future patients, we calculated a 95% prediction interval(prI).

### Results

Thirteen trials were included, involving 624 participants. Pooled results showed a significant improvement in HF for immediate (SMD = 1.01, 95% CI: 0.44 to 1.59) and short-term effects (SMD = 1.21, 95% CI: 0.90 to 1.52). Subgroup analyses by type of comparison group showed that these techniques are more effective than the control group in the immediate and short term and than static stretching in the short term. Analyses by total sessions showed a significant increase in HF with a treatment of 1, 3, 10 and 12 sessions.

**Data Availability Statement:** All relevant data are within the manuscript and its Supporting information files.

**Funding:** This work was supported by FEDER funds. S-NDAA (2020-PREDUCLM-16704), S-RG (2020-PREDUCLM-15596) and B-BP (2020-PREDUCLM-16746) are supported by a grant from the University of Castilla-La Mancha.

**Competing interests:** The authors have declared that no competing interests exist.

## Conclusion

Neurodynamic techniques improve HF immediately and in the short term. Subgroup analyses by type of comparison group showed that these techniques are more effective than static stretching in the short term.

## 1. Introduction

The hamstring muscles have a high tendency to shorten due to their multijoint condition, their postural tonic nature, and the considerable tensile forces to which they are constantly subjected [1]. In sports performance, athletes with good hamstring flexibility (HF) have better performance scores in acceleration, sprinting, jumping and agility tests [2]. Furthermore, an adequate HF is a key factor in preventing muscle injuries, patellar tendinopathy, muscle imbalances, patellofemoral pain, hamstring syndrome and low back pain [1], saving elite teams an estimated €500,000 per month [3].

Due to the importance of good HF for sports performance and health and the economic impact of injuries on the healthcare system and sports teams, understanding how to improve the flexibility of this muscle group has become a key objective for researchers, trainers, coaches and physiotherapists. Static stretches are the most common and well-known exercises for improving flexibility. However, techniques such as dynamic stretching, proprioceptive neuromuscular facilitation, whole-body vibration, low-frequency electrical stimulation, Pilates, ballistic stretching, eccentric exercises or neurodynamic techniques have shown promising results in improving these parameters [4].

Neurodynamic interventions are physiotherapy methods based on exercises or manual therapy that mobilize peripheral nerves or adopt postures that unload the nervous system to facilitate movement between the nervous system and its interfaces (surrounding structures) and that decrease the mechanical load on the nervous system to tolerate the traction, compression and friction forces involved in daily activities and sport [5,6]. Muscle flexibility depends not only on the properties of the muscle (such as viscoelasticity or the number of serial sarcomeres) but also on the subject's "stretch tolerance" and its relationship to the adjacent connective and nervous tissue [7]. In this case, abnormal sciatic nerve mechanosensitivity has been shown to cause poor HF [8]. Therefore, neurodynamic techniques could play a role in HF.

A previous systematic review [9] suggested that neurodynamic techniques could be useful in improving HF. However, when neurodynamic techniques were compared with other treatments (static stretching, suboccipital muscle inhibition, PNF. . .), these treatments were mixed in the same analysis, which prevented us from knowing their independent effect. Moreover, since then, some randomized controlled trials on the efficacy of neurodynamic techniques to improve HF have been published. Thus, the aims of this systematic review with meta-analysis were (i) to estimate the effect of neurodynamic techniques on HF and (ii) to compare the effect of these techniques with static stretching.

## 2. Materials and methods

Our systematic review with meta-analysis (PROSPERO registration number: CRD42022349138) was conducted following the Preferred Reporting Items for Systematic Reviews and Meta-Analyses (PRISMA-S and PRISMA) [10] guidelines (S2 File) and reported according to the recommendations of the Cochrane Collaboration Handbook [11].

## 2.1. Databases and search strategy

We searched the Cochrane, Pubmed, Scopus, WOS and Sportdiscus databases from the date of their inception to May 22$^{nd}$, 2024, with no language restrictions. We used the following controlled and free-text terms: "athletic performance", performance, "sport performance", flexibility, extensibility, ROM, "nerve treatment", "nerve therapy", "neural mobilization", neurodynamic, neurodynamics, "nerve tension", "nerve stretch", "neural tension", "nerve mobilization", "neural glide", "neural treatment", "nerve gliding", nerve glide, "neural gliding", "nerve gliding exercises", "neuromobilization maneuver", neuromobilization, "neurodynamic techniques", RCT, "randomized clinical trial", "randomized trial" and "randomized controlled trial". The search strategy used for each database is presented in S1 Table in S1 File of the supplementary material. In addition, we manually searched the references of the included articles to identify additional eligible studies. In addition, we supplemented the electronic search with manual searches for additional randomized controlled trials (RCTs) by reviewing the reference lists of previous reviews and with manual searches for published, unpublished, and ongoing RCTs in international trial registers such as ClinicalTrials.gov [12]. For example, we searched ClinicalTrials.gov using the search term "hamstring flexibility" combined with neural mobilization.

## 2.2. Eligibility criteria

We included: (i) randomized clinical trials published in any language, (ii) comparing neurodynamic interventions with control intervention (sham, no intervention or the same protocol treatment of intervention group except neurodynamic mobilizations) or with static stretching exercises, (iii) for HF (measured using tests such as the knee extension angle (KEA) and the straight leg raise (SLR)), in (iv) healthy asymptomatic subjects with limited HF. Finally, we excluded (i) trials comparing different protocols of neurodynamic interventions and (ii) review articles, guidelines, editorials, case reports and comments.

Two reviewers independently conducted the literature search, screening and trial selection (S-NDAA and B-BP). A third researcher resolved disagreements (V-MV).

## 2.3. Data synthesis and outcome definition

After pilot testing of data extraction in June 2024, independent pairs of reviewers extracted the following information from each study: (i) author information and year of publication; (ii) characteristics of the sample (country, sample size, and age distribution); (iii) intervention characteristics (control and intervention regimen, duration and frequency of interventions); and (iv) outcomes measured (scale, basal and post-intervention values). Two reviewers extracted data independently (S-NDAA and A-TC). Disagreements were resolved by a third researcher (V-MV). We contacted trial authors up to three times if necessary to obtain missing information.

Our primary outcome was the efficacy (response measured by different tests) of neurodynamic interventions for improving HF. For analyses, we recorded the outcome at the end of one session (immediate effects) and at the end of the entire intervention period (short-term effects).

## 2.4. Risk of bias assessment

In accordance with the Cochrane Handbook for Systematic Reviews of Intervention, we used the revised Cochrane Collaboration´s Risk of Bias 2 tool (RoB 2.0) [13] to assess the risk of bias according to five domains: randomization process, deviations from intended

interventions, missing outcome data, measurement of the outcome, and selection of the reported result. Overall, we determined to have a "low risk of bias" if all domains were "low risk," to have "some concerns" if at least one domain was designated as "some concerns" and no domains were identified "high risk," and to have a "high risk" if at least one domain was "high risk".

The quality assessment was performed independently by two reviewers (S-NDAA and S-RG), and inconsistencies were resolved by consensus or involving a third researcher (V-MV).

## 2.5. Measures of intervention effect

We calculated the standardized mean differences (SMDs) with their 95% confidence intervals (CIs) using Cohen´s d index as the SMD method [14]. Positive SMD values indicate an improvement in HF in favor of the neurodynamic intervention group. Cohen's d values of approximately 0.2 indicate weak effects, 0.5 moderate effects, 0.8 strong effects, and above 1.0 very strong effects.

## 2.6. Data synthesis and statistical analysis

We conducted a random-effects meta-analysis using the DerSimonian and Laird method [15]. We employed subgroup analyses according to the type of comparison group (control group or static stretching exercises) and total number of sessions. We also calculated a 95% prediction interval (prI) to reflect the variation in true treatment effects in different settings, including future studies, to aid clinical decision-making [16].

We examined the statistical heterogeneity in each comparison by visual inspection of the forest plot. We also calculated the $I^2$ statistic, whose values were considered not important (0–40%), moderate (30–60%), substantial (50–90%), or considerable (75–100%), and corresponding p values were also considered [11].

We performed sensitivity analyses using the leave-one-out method [11] to assess the robustness of the summary estimates, and we assessed publication bias by Egger´s regression asymmetry test [17]. P values <0.10 were considered statistically significant.

We conducted all analyses in R (version 4.4.0) using the R package "meta" and the datasets generated during and/or analyzed during the current study are available in supplementary material S1 Data.

# 3. Results

## 3.1. Systematic review

After removing duplicates, the search identified 148 citations (S3 File). Of them 22 were considered as potentially eligible and examined in full text (Fig 1). Finally, we included 13 trials [7,18–29], including 624 participants aged between 17 and 68.4 years, and conducted on three continents (six in Asia, five in Europe, one in North America and one in South America) between 1997 and 2023. The excluded studies with reasons are available in S2 Table in S1 File.

Neurodynamic techniques were compared with a control group in 8 studies and with static stretching exercises in 7 studies. The length of the intervention ranged from 1 to 6 weeks with a frequency between 1 and 7 sessions per week. Concerning the different tests used, HF was measured using the knee extension angle (KEA) and the straight leg raise test (SLR). The characteristics of the included studies and the baseline values are shown in Table 1.

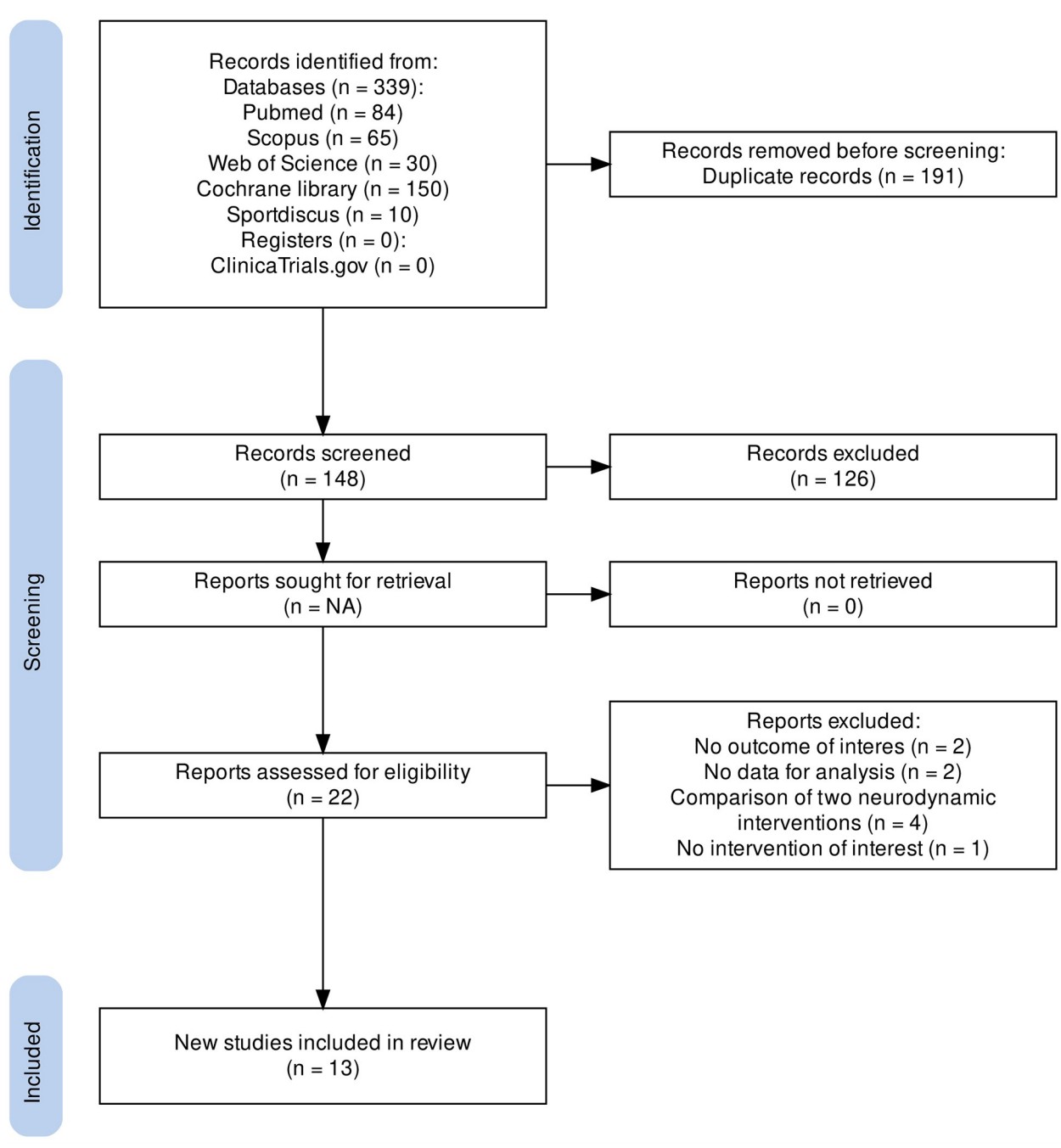

**Fig 1. PRISMA flowchart.**

**Table 1. Characteristics of the studies included in the meta-analysis.**

| STUDY CHARACTERISTICS | | POPULATION CHARACTERISTICS | | INTERVENTION CHARACTERISTICS | | | OUTCOME | | |
|---|---|---|---|---|---|---|---|---|---|
| Study, year | Country | n (% female) | Age (years) Mean (SD) | Groups by intervention | Duration (weeks) | Frequency (x/week) | Scale | Basal values | Post-intervention values |
| Alshammari, 2019 [18] | Jordan | IG1: 20 (NR) IG2: 20 (NR) IG3: 20 (NR) | IG1: 21.5 (0.9) IG2: 21.5 (0.6) | IG1: Neurodynamic techniques + static stretching IG2: Static stretching | 1 | 1 | KEA | NR | NR |
| Areeudomwong, 2016 [19] | Thailand | IG: 20 (0) CG: 20 (0) | IG: 19.82 (1.40) CG: 20.09 (1.14) | IG: Neurodynamic techniques CG: Control group | 4 | 3 | KEA | IG: 23.58 (3.78) CG: 24.83 (2.76) | IG: 11.67 (6.8) CG: 23.42 (2.61) |
| Castellote-Caballero, 2013 [20] | Spain | IG: 14 (0) CG: 14 (0) | IG: 20.79 (1.05) CG: 20.71 (0.99) | IG: Neurodynamic techniques CG: Control group | 1 | 3 | SLR | IG1: 58.1 CG: 58.9 | IG1: 67.4 CG: 59.1 |
| Castellote-Caballero, 2014 [21] | Spain | IG1: 40 (50) IG2: 40 (50) CG: 40 (50) | IG1: 33.7 (7.68) IG2: 33.9 (7.44) CG: 32.7 (7.08) | IG1: Neurodynamic techniques IG2: Static stretching CG: Control group | 1 | 1 | SLR | IG1: 59.8 (4.70) IG2: 59.9 (6.99) CG: 59.4 (5.68) | IG1: 69.7 (3.69) IG2: 65.5 (7.97) CG: 59.4 (5.45) |
| De Ridder, 2019 [7] | Belgium | IG1: 25 (0) IG2: 25 (0) | IG1: 23.4 (2.02) IG2: 21.8 (2.39) | IG1: Neurodynamic techniques IG2: Static stretching | 6 | Daily | SLR | IG1: 57.5 (7.57) IG2: 57.6 (4.98) | IG1: 70.1 (6.22) IG2: 66.9 (3.98) |
| Escobar, 2023 [29] | Peru | IG1: 8 (NR) CG: 8 (NR) | IG1: 20.7 (1.7) CG: 20.3 (1.7) | IG1: Neurodynamic techniques CG: Control group | 1 | 1 | SLR | IG1: 58.67 (9.24) CG: 64 (6.44) | Immediate term IG1: 64.5 (6.43) CG: 64.75 (6.88) Short-term IG1: 77.88 (9.6) CG: 66.75 (6.77) |
| Guru-Karthick, 2019 [22] | India | IG1: 35 IG2: 35 CG: 35 | 17–25 | IG1: Neurodynamic techniques IG2: Neurodynamic techniques CG: Control group | 1 | 1 | KEA | IG1: 55.32 (12.14) IG2: 61.20 (10.83) CG: 58.12 (8.11) | IG1: 63.8 (11.93) IG2: 74.96 (9.80) CG: 58.56 (8.05) |
| Mendez-Sanchez, 2010 [23] | Spain | IG1: 4 (0) IG2: 4 (0) | 21 (3) | IG1: Neurodynamic techniques + static stretching IG2: Static stretching | 1 | 1 | SLR | IG1: 85.13 (2.08) IG2: 85.10 (4.16) | IG1: 91.25 IG2: 88.0 |
| Mhatre, 2013 [24] | India | IG1: 28(0) IG2: 28(0) | IG1: 22.32 (1.84) IG2: 21.46 (2.50) | IG1: Neurodynamic techniques IG2: Static stretching | 1 | 1 | KEA | IG1: 31.78 (8.18) IG2: 34.64 (6.65) | IG1: 21.96 (8.20) IG2: 24.64 (9.22) |
| Muragod, 2017 [25] | India | IG1: 10 (NR) IG2: 10 (NR) | IG1: 66.2 (1.322) IG2: 68.4 (3.59) | IG1: Neurodynamic techniques IG2: Static stretching | 2 | 5 | SLR | IG1: 57.40 (2.47) IG2: 59.05 (6.86) | IG1: 63.90 (2.8) IG2: 61.65 (6.74) |
| Satkinskiene, 2019 [26] | Lithuania | IG1: 11 (NR) IG2: 11 (NR) | IG1: 22 (1.9) IG2: 22 (2.0) | IG1: Neurodyamic techniques IG2: Static stretching | 1 | 1 | SLR | NR | NR |

(*Continued*)

**Table 1.** (Continued)

| STUDY CHARACTERISTICS | | POPULATION CHARACTERISTICS | | INTERVENTION CHARACTERISTICS | | | OUTCOME | | |
|---|---|---|---|---|---|---|---|---|---|
| Study, year | Country | n (% female) | Age (years) Mean (SD) | Groups by intervention | Duration (weeks) | Frequency (x/week) | Scale | Basal values | Post-intervention values |
| Sharma, 2016 [27] | India | IG1: 20 (NR) IG2: 20 (NR) CG: 20 (NR) | IG1: 21.75 (1.86) IG2: 21.70 (2.25) CG: 22.80 (2.64) | IG1: Neurodynamic techniques + static stretching IG2: Neurodynamic techniques + static stretching CG: Static stretching | 1 | 3 | KEA | IG1: 33.3 (6.9) IG2: 30.8 (6.8) CG: 32.8 (8.5) | IG1: 14.7 (6.5) IG2: 9.9 (7.2) CG: 21.5 (12.3) |
| Webright, 1997 [28] | USA | IG1: 11 (45.45) IG2: 14 (64.28) IG3: 14 (36.36) | IG1: 20.8 (3.16) IG2: 21.2 (3.65) CG: 22.1 (2.50) | IG1: Neurodynamic techniques IG2: Static stretching CG: Control group | 6 | Daily | KEA | IG1: 34 (7.7) IG2: 35.1 (9.7) CG: 31.4 (6.6) | IG1: 23.8 (10.4) IG2: 26.2 (9.2) CG: 33.9 (6.9) |

Values are presented as mean ± SD unless otherwise indicated.

**Abbreviations**: CG: Control group; KEA: Knee extension angle; IG: Intervention group; NR: Not reported; SD: Standard deviation; SLR: Straight Leg Raise.

### 3.2. Risk of bias assessment

For immediate effects (S1 Fig-A in S1 File), seven studies showed a moderate risk of bias. In particular, all included studies had shortcomings in the selection of the reported results, and in the randomization process.

For short-term effects (S1 Fig-B in S1 File), six studies had a moderate risk of bias, and one a low risk of bias. Six of the studies had shortcomings in the selection of reported results, and one in the randomization process.

**The** S4 File contains a comprehensive report that addresses each question of the RoB 2.0 tool from each of the included studies for each outcome.

### 3.3. Meta-analysis

**3.3.1. Immediate effects.** Seven studies (9 samples) assessing immediate effects were included in the meta-analysis (Fig 2). Pooled results showed a significant improvement in HF (SMD = 1.01, 95% CI: 0.44 to 1.59) with substantial heterogeneity ($I^2$ = 86%, p<0.01). Analyses by type of comparison group showed significantly more HF in the neurodynamic group than in the control group (SMD = 1.28, 95% CI: 0.59 to 1.96) but not significantly when compared to static stretching exercises (SMD = 0.47, 95% CI: -0.12 to 1.06). The prI was not significant for any subgroup.

**3.3.2. Short-term effects.** Seven studies (9 samples) assessing the short-term effects were included in the meta-analysis (Fig 3). Pooled results showed a significant improvement in HF (SMD = 1.21, 95% CI: 0.90 to 1.52) with a significant 95% prI (0.51 to 1.92) and no important heterogeneity ($I^2$ = 29%, p = 0.19). Analyses by the type of comparison group showed significantly more HF in the neurodynamic group than in the control group (SMD = 1.35, 95% CI: 1.02 to 1.67) with a significant prI (0.89 to 1.80) and significantly more HF compared to static stretching exercises (SMD = 0.86, 95% CI: 0.13 to 1.59).

**3.3.3. Dose–response analysis.** Analyses by total sessions showed a significant increase in HF with a treatment of 1 session (9 studies; SMD = 1.03, 95% CI: 0.46 to 1.60), 3 sessions (3 studies; SMD = 1.18, 95% CI: 0.83 to 1.53), 10 sessions (1 study; SMD = 3.25, 95% CI: 1.83 to 4.68)

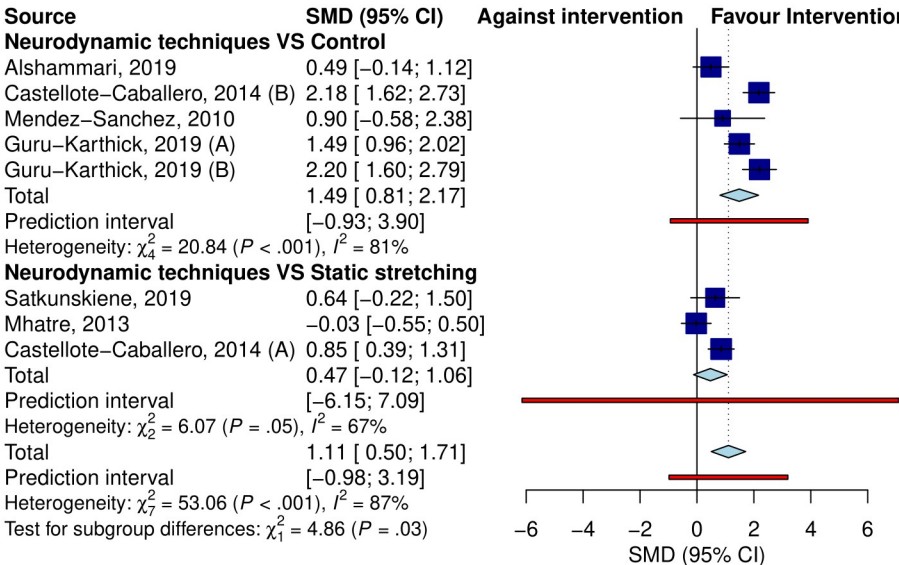

**Fig 2. Forest plot of the pooled effect of neurodynamic techniques compared to the control group and compared to static stretching in the immediate term.**

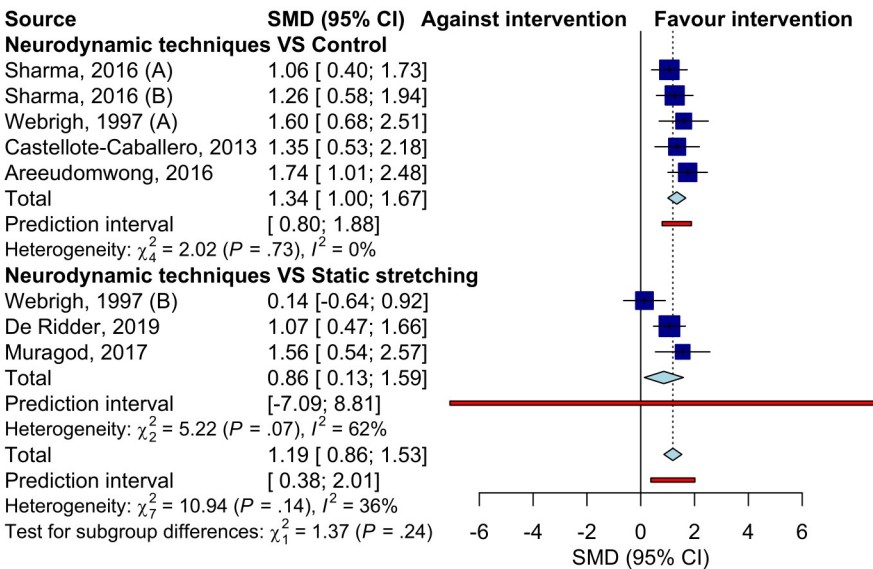

**Fig 3. Forest plot of the pooled effect of neurodynamic techniques compared to the control group and compared to static stretching at short term.**

and 12 sessions (1 study; SMD = 1.71, 95% CI: 0.97 to 2.44). However, nonsignificant results showed treatments with 42 sessions (3 studies; SMD = 0.89, 95% CI: -0.83 to 2.61) (Fig 4).

## 3.4. Sensitivity analysis

When one-by-one studies were removed from the analysis to examine the impact of individual studies, the pooled SMD estimates for neurodynamic techniques on HF did not change significantly on immediate and short-term effects (S2 and S3 Figs in S1 File).

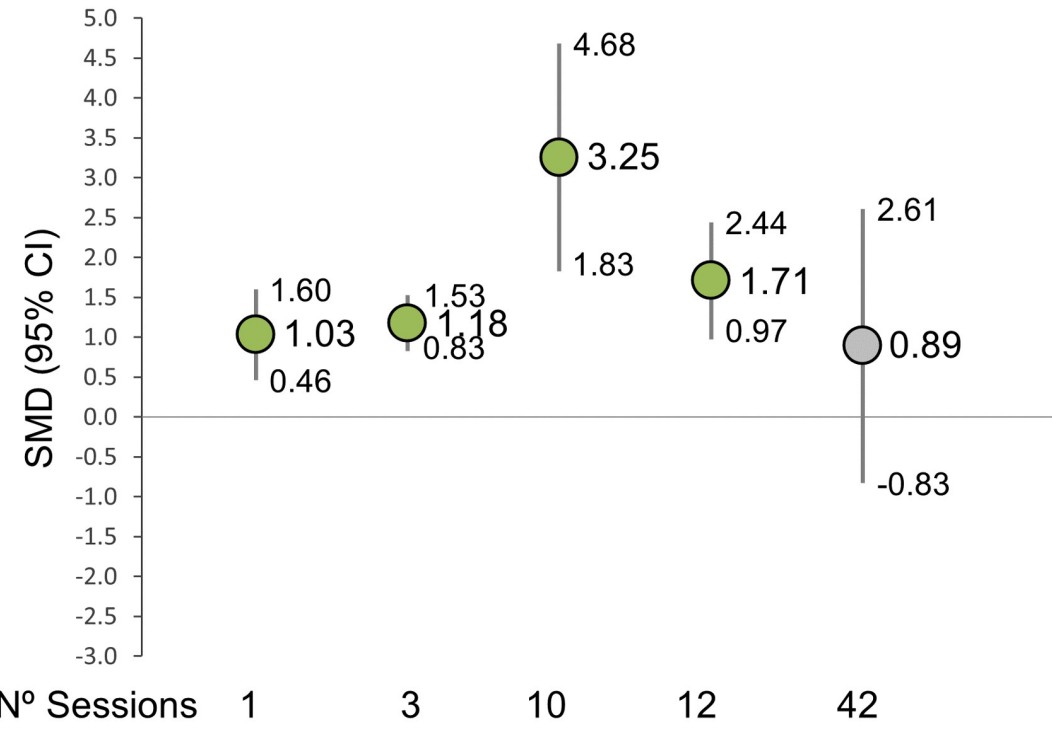

**Fig 4. Dose response analysis by total number of sessions.**

### 3.5. Publication bias

There was no evidence of publication bias on immediate and short-term effects, as shown by visual inspection of the funnel plot (S4 and S5 Figs in S1 File) and Egger´s tests (p = 0.72 and p = 0.43, respectively).

## 4. Discussion

This systematic review with meta-analysis provides a synthesis of evidence suggesting that neurodynamic techniques are a suitable intervention to improve HF in asymptomatic subjects immediately and in the short term. In addition, subgroup analyses showed that these techniques, in the short term, are more effective than static stretching. Furthermore, our analyses support that neurodynamic techniques are effective interventions from the first session.

A previous systematic review with meta-analysis [9] concluded that neurodynamic techniques could be considered more effective for HF than nonintervention and other physical therapy treatments. However, it is not possible to know against which methods neurodynamic techniques are more effective in improving HF because it did not separately examine the effect of each available modality. This systematic review and meta-analysis reinforces some of the conclusions of this previous study (where these techniques were compared with a control group), including six subsequent randomized controlled trials, and shows statistically significant differences in favor of neurodynamic techniques when compared to static stretching, which is the best known and most widely used approach for the improvement of HF in elite sports and in the general population. In addition, a dose–response analysis was performed to determine the effect of these interventions considering the total number of intervention sessions.

Neurodynamic techniques can be classified into two categories: gliding techniques and tensioning techniques. Of the studies included in this review, 11 applied a protocol consisting of gliding techniques (10 focused on the sciatic nerve [19–23,25–27,29] and 1 on the tibial nerve [18]), 3 applied nerve tensioning techniques to the sciatic nerve [22,27,28], and 1 study applied the Mulligan Ben Leg Raise technique [24]. It is also noteworthy that all studies applied neurodynamic techniques in isolation, with only three studies combining them with static stretching techniques [18,23,27]. Regarding the dose-response analysis, it is interesting to note that intervention protocols with a total of 42 sessions do not obtain statistically significant improvements, while protocols with fewer sessions (such as 3, 10 or 12) do find this statistical significance. This discrepancy could be attributed to the fact that these protocols encompass the highest number of total sessions and the highest number of weekly sessions (six sessions per week). This suggests the possibility of adverse effects, such as nervous irritation, which may occur with certain frequencies of sessions. However, further research is necessary to draw definitive conclusions on this matter.

There is moderate evidence that HF is associated with musculoskeletal injury [30], and static stretching is the best known and most commonly used technique to improve HF due to the change in the mechanical or viscoelastic properties of the muscle [31]. However, joint movement during static stretching exercises causes mechanical stress on peripheral nerves caused by adhesions between the epineurium and surrounding tissues that prevent the nerves from moving freely, increasing central nervous system alertness and provoking pain [26,32]. Furthermore, the acute effects of static stretching are associated with decreased maximum muscular strength and power [33], which may not be advisable for sports practice. The improvement in HF following neurodynamic techniques may be due to the modification of the adherences between the neural tissue and the interfaces, decreasing neural mechanosensitivity and increasing the viscoelasticity of the neural tissue [8,34,35]. In addition, these techniques could modify the subject's sensation of pain by producing an analgesic effect and, therefore, the associated protective muscle contraction. Within the domain of rehabilitation, though with limited evidence [36–38], it appears that the efficacy of these techniques is sustained in patients with conditions such as lumbar radiculopathy, enhancing the flexibility of the hamstrings and the mobility of the knee and hip.

## 4.1 Limitations

First, although the design of all included trials allowed us to calculate estimates of the effects of neurodynamic techniques, in some trials, neurodynamic techniques were included as part of a combined treatment with static stretching. Second, because of the scarcity of trials, we were unable to perform a subgroup analysis by type of technique (tension or sliding). Third, the risk of bias of the included trials was mainly moderate, and some concerns, such as the selection of the reported results, should not be ignored. Fourth, the design of trials was heterogeneous in terms of the type of intervention. Fifth, the included trials examined the short-term effects of neurodynamic techniques (completely after the end of the intervention regime). Therefore, future research should examine the effect of these techniques in the mid- and long-term.

## 5. Conclusions

Neurodynamic techniques improve HF in asymptomatic healthy subjects immediately and in the short term. Subgroup analyses by type of comparison group showed that these techniques are more effective than static stretching in the short term. This finding has important implications because they endorse the use of neurodynamics as first-choice techniques for HF. It remains to be elucidated whether neurodynamic therapy together with stretching increases

improvements in HF or not, as well as to compare neurodynamic therapy with other techniques to improve HF.

## Supporting information

**S1 Data. Datasets generated during and analyzed during the current study.**
(XLSX)

**S1 File. Supplementary tables and figures.**
(DOCX)

**S2 File. PRISMA checklist.**
(DOCX)

**S3 File. Articles screened after removing duplicates.**
(XLSX)

**S4 File. Comprehensive report of the Cochrane risk-of-bias tool for randomized trials.**
(XLSX)

## Author Contributions

**Conceptualization:** Sergio Núñez de Arenas-Arroyo, Vicente Martínez-Vizcaíno, Sara Reina-Gutiérrez, Bruno Bizzozero-Peroni, Iván Cavero-Redondo.

**Data curation:** Sergio Núñez de Arenas-Arroyo, Vicente Martínez-Vizcaíno, Ana Torres-Costoso, Sara Reina-Gutiérrez, Iván Cavero-Redondo.

**Formal analysis:** Sergio Núñez de Arenas-Arroyo, Bruno Bizzozero-Peroni.

**Methodology:** Sergio Núñez de Arenas-Arroyo, Ana Torres-Costoso, Bruno Bizzozero-Peroni.

**Supervision:** Vicente Martínez-Vizcaíno, Iván Cavero-Redondo.

**Writing – original draft:** Sergio Núñez de Arenas-Arroyo.

**Writing – review & editing:** Sergio Núñez de Arenas-Arroyo, Vicente Martínez-Vizcaíno, Ana Torres-Costoso, Sara Reina-Gutiérrez, Bruno Bizzozero-Peroni, Iván Cavero-Redondo.

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
