## [Decision Letter · Decision Letter 0]

23 Dec 2024

PONE-D-24-26834Immediate and short-term effects of neurodynamic techniques on hamstring flexibility: A systematic review with meta-analysis.PLOS ONE

Dear Dr. Martínez-Vizcaíno,

Thank you for submitting your manuscript to PLOS ONE. After careful consideration, we feel that it has merit but does not fully meet PLOS ONE’s publication criteria as it currently stands. Therefore, we invite you to submit a revised version of the manuscript that addresses the points raised during the review process.

**ACADEMIC EDITOR: **

We apologise for the incovenience about the deadline of review schedule. Nevertheless, we have invited more than 30 reviewers without any response. Currently, we have secured 2 reviewers with a great effort. Thus, according to reviewers comments, a decision of minor revision has been provided. Please, respond to all reviewers comments in an accurate manner as soon as possible to avoid more delay in our final decision. Thanks.

We look forward to receiving your revised manuscript.

Kind regards,

César Calvo-Lobo, PhD, MSc, PT

Academic Editor

PLOS ONE

Journal requirements: When submitting your revision, we need you to address these additional requirements. 1. Please ensure that your manuscript meets PLOS ONE's style requirements, including those for file naming. The PLOS ONE style templates can be found at https://journals.plos.org/plosone/s/file?id=wjVg/PLOSOne_formatting_sample_main_body.pdf and https://journals.plos.org/plosone/s/file?id=ba62/PLOSOne_formatting_sample_title_authors_affiliations.pdf. 2. We noticed you have some minor occurrence of overlapping text with the following previous publication(s), which needs to be addressed: https://pubmed.ncbi.nlm.nih.gov/36361353/ In your revision ensure you cite all your sources (including your own works), and quote or rephrase any duplicated text outside the methods section. Further consideration is dependent on these concerns being addressed. 3. We note that the grant information you provided in the ‘Funding Information’ and ‘Financial Disclosure’ sections do not match.  When you resubmit, please ensure that you provide the correct grant numbers for the awards you received for your study in the ‘Funding Information’ section. 4. In the online submission form, you indicated that [The datasets generated during and/or analyzed during the current study are not publicly available, but are available from the corresponding author on reasonable request.]. All PLOS journals now require all data underlying the findings described in their manuscript to be freely available to other researchers, either 1. In a public repository, 2. Within the manuscript itself, or 3. Uploaded as supplementary information.This policy applies to all data except where public deposition would breach compliance with the protocol approved by your research ethics board. If your data cannot be made publicly available for ethical or legal reasons (e.g., public availability would compromise patient privacy), please explain your reasons on resubmission and your exemption request will be escalated for approval.  5. As required by our policy on Data Availability, please ensure your manuscript or supplementary information includes the following:  A numbered table of all studies identified in the literature search, including those that were excluded from the analyses.   For every excluded study, the table should list the reason(s) for exclusion.   If any of the included studies are unpublished, include a link (URL) to the primary source or detailed information about how the content can be accessed.  A table of all data extracted from the primary research sources for the systematic review and/or meta-analysis. The table must include the following information for each study:  Name of data extractors and date of data extraction  Confirmation that the study was eligible to be included in the review.   All data extracted from each study for the reported systematic review and/or meta-analysis that would be needed to replicate your analyses.  If data or supporting information were obtained from another source (e.g. correspondence with the author of the original research article), please provide the source of data and dates on which the data/information were obtained by your research group.  If applicable for your analysis, a table showing the completed risk of bias and quality/certainty assessments for each study or outcome.  Please ensure this is provided for each domain or parameter assessed. For example, if you used the Cochrane risk-of-bias tool for randomized trials, provide answers to each of the signalling questions for each study. If you used GRADE to assess certainty of evidence, provide judgements about each of the quality of evidence factor. This should be provided for each outcome.   An explanation of how missing data were handled.   This information can be included in the main text, supplementary information, or relevant data repository. Please note that providing these underlying data is a requirement for publication in this journal, and if these data are not provided your manuscript might be rejected.  

Additional Editor Comments:

We apologise for the incovenience about the deadline of review schedule. Nevertheless, we have invited more than 30 reviewers without any response. Currently, we have secured 2 reviewers with a great effort. Thus, according to reviewers comments, a decision of minor revision has been provided. please, respond to all reviewers comments in an accurate manner as soon as possible to avoid more delay in our final decision. Thanks.

Reviewers' comments:

Reviewer's Responses to Questions

**Comments to the Author**

1. Is the manuscript technically sound, and do the data support the conclusions?

Reviewer #1: Yes

Reviewer #2: Yes

2. Has the statistical analysis been performed appropriately and rigorously? 

Reviewer #1: Yes

Reviewer #2: Yes

3. Have the authors made all data underlying the findings in their manuscript fully available?

Reviewer #1: Yes

Reviewer #2: Yes

4. Is the manuscript presented in an intelligible fashion and written in standard English?

Reviewer #1: Yes

Reviewer #2: Yes

5. Review Comments to the Author

Reviewer #1: This paper is a review with meta-analysis focused on the comparison of neurodynamic techniques versus static stretching for the improvement of hamstring flexibility. In genera, I consider that the review is in conformity with the Plos One acceptance criteria. I highlight the methodology used as a strong point of the review. Two reviewers (plus a third in case of discrepancy) are included to carry out the literature search, data extraction and risk of bias assessment. The statistical analysis of the meta-analysis is very adequate. However, some aspects should be considered.

In my opinion, this study is not very innovative in terms of comparing neurodynamics with static stretching. Although it is important that, compared to previous reviews, it analyses more RCTs and adds relevant assessments such as the dose-response relationship.

I think the criteria for the selection of studies are a bit too brief. Some more could be added, such as assessment tests to define hamstring flexibility.

I consider that the search strategy used in each of the databases should be added.

Inclusion criteria indicate that the sample consists of adults (line 116), although one of the studies includes subjects aged 17 (line 171). Although it is only one year difference, I think that the inclusion criteria should be corrected.

I think it would be appropriate to set a specific time period to define the short-term effects (lines 132-133).

In order to keep the text uniform, ‘’hamstring flexibility`` could be changed to HF (line 213).

In statistical methodology it is described that Cohen's d is used to calculate the effect size. But the results do not express this statistic.

In my opinion, the methodology or discussion should describe what kind of neurodynamic techniques are performed in the studies included in the review; or whether the neurodynamic and stretching treatment is isolated or combined with other interventions.

The discussion seems to be limited. It could be expanded commenting on some of the results found in the review, such as: the difference found in healthy people could vary in people with pathology; talk about the dose-response relationship (for example, it is curious that in a study of 42 sessions there is no significant improvement, while in others with many fewer sessions there are significant changes-line 221); there are two studies included in the review with very small sample sizes, which should be interpreted with caution; comparison of results with other similar reviews.

Reviewer #2: In general terms, the manuscript provides solid and detailed information on the research process. The intervention characteristics of the trials included in the review are of great relevance. Such as the duration and frequency of the interventions.

The manuscript provides sufficient information and data to allow each reader to draw his or her own conclusions, the main objective being the knowledge of the effects, in isolation, of neurodynamic techniques on hamstring flexibility. Furthermore, it is quite correct to offer conclusions based on the analyses carried out by subgroups and not on global analyses. Likewise, in the selection of articles included in the review, we rejected those that combined neurodynamic techniques with other stretching techniques.

However, with regard to the subjacent data, I would like to point out that it would be of great help to the reader to include in Table 1 of the article the pre and post angular measurements of the KEA and SLR exploratory maneuvers. And not only show the initial measurement at the price of intervention.

As for the statistical analysis I have nothing to add.

Translated with DeepL.com (free version)

6. PLOS authors have the option to publish the peer review history of their article (what does this mean?). If published, this will include your full peer review and any attached files.

Reviewer #1: No

Reviewer #2: No

---

## [Author Response · Author response to Decision Letter 0]

8 Jan 2025

César Calvo-Lobo and Emily Chenette,

Academic Editor and Editor-in-Chief 

PLOS ONE

Enclosed you will find a revision of our manuscript: “Immediate and short-term effects of neurodynamic techniques on hamstring flexibility: A systematic review with meta-analysis.”

We would like to thank you for giving us the opportunity to revise and improve our manuscript; we also thank the reviewers for their thoughtful and constructive comments. 

We have considered all of the suggestions and have incorporated them into the revised manuscript. Changes to the original manuscript are shown in red font, and we believe our manuscript is stronger as a result of these modifications. An itemized point-by-point response to the editor and reviewers’ comments is presented below.

Vicente Martinez-Vizcaino.

Universidad de Castilla-La Mancha

Edificio Melchor Cano, Centro de Estudios Socio-Sanitarios

Santa Teresa Jornet s/n, 16071 Cuenca, Spain.

E-mail: Vicente.Martinez@uclm.es

JOURNAL REQUIREMENTS:

1. Please ensure that your manuscript meets PLOS ONE's style requirements, including those for file naming. The PLOS ONE style templates can be found at https://journals.plos.org/plosone/s/file?id=wjVg/PLOSOne_formatting_sample_main_body.pdf and https://journals.plos.org/plosone/s/file?id=ba62/PLOSOne_formatting_sample_title_authors_affiliations.pdf. 

Authors: Done

https://pubmed.ncbi.nlm.nih.gov/36361353/

In your revision ensure you cite all your sources (including your own works), and quote or rephrase any duplicated text outside the methods section. Further consideration is dependent on these concerns being addressed.

Authors: We have reworded any duplicate text from the previous publication to address the overlap.

Authors: Done

4. In the online submission form, you indicated that [The datasets generated during and/or analyzed during the current study are not publicly available, but are available from the corresponding author on reasonable request.]. 

Authors: The task has been completed. The following sentences have been incorporated into the main text, and the S1 data file has been uploaded with the database.

Data Availability: The datasets generated during and/or analyzed during the current study are available in the supplementary material S1 data.

5. As required by our policy on Data Availability, please ensure your manuscript or supplementary information includes the following: 

Authors: Done. It is shown in S2 file (total articles screened), Figure 1 (flow chart), Table 1 (Characteristics of the studies included in the meta-analysis) and S2 table (Studies excluded after full text read with the reason for exclusion).

Furthermore, we have added the following lines in the main text: 

“After removing duplicates, the search identified 148 citations (S2 file). Of them 22 were considered as potentially eligible and examined in full text (Fig 1). Finally, we included 13 trials[7,18,27–29,19–26], including 624 participants aged between 17 and 68.4 years, and conducted on three continents (six in Asia, five in Europe, one in North America and one in South America) between 1997 and 2023. The excluded studies with reasons are available in S2 Table.”

Authors: Excluded studies with reasons are available in S2 Table.

Authors: Not applicable.

Name of data extractors and date of data extraction.

Authors: Done.

Data synthesis and outcome definition section:

“After pilot testing of data extraction in June 2024, independent pairs of reviewers extracted the following information from each study: (i) author information and year of publication; (ii) characteristics of the sample (country, sample size, and age distribution); (iii) intervention characteristics (control and intervention regimen, duration and frequency of interventions); and (iv) outcomes measured (scale, basal and post-intervention values). Two reviewers extracted data independently (S-NDAA and A-TC). Disagreements were resolved by a third researcher (V-MV). We contacted trial authors up to three times if necessary to obtain missing information.”

Authors: Thank you. We have added all data extracted from each study in Table 1 and S1 data.

Authors: Not applicable. No information were obtained from another source.

Authors: A supplementary file (S3 file) containing the comprehensive report of the Cochrane risk-of-bias tool for randomized trials for each outcome has been uploaded. In addition, the S1 figure has been modified, and the following sentences have been added to the risk of bias results section:

3.2. Risk of bias assessment

“For immediate effects (S1 Fig-A), seven studies showed a moderate risk of bias. In particular, all included studies had shortcomings in the selection of the reported results, and in the randomization process.

For short-term effects (S1 Fig-B), six studies had a moderate risk of bias, and one a low risk of bias. Six of the studies had shortcomings in the selection of reported results, and one in the randomization process.

The S3 file contains a comprehensive report that addresses each question of the RoB 2.0 tool from each of the included studies for each outcome.”

Authors: Not applicable

Authors: Done. All this information was included in the main text or in supplementary information.

Authors: Done.

ADDITIONAL EDITOR COMMENTS:

We apologise for the incovenience about the deadline of review schedule. Nevertheless, we have invited more than 30 reviewers without any response. Currently, we have secured 2 reviewers with a great effort. Thus, according to reviewers comments, a decision of minor revision has been provided. please, respond to all reviewers comments in an accurate manner as soon as possible to avoid more delay in our final decision. Thanks.

Authors: Thank you for the editor´s comment. We have addressed all of the comments and have made improvements to the quality of the manuscript.

REVIEWERS' COMMENTS:

Reviewer #1: This paper is a review with meta-analysis focused on the comparison of neurodynamic techniques versus static stretching for the improvement of hamstring flexibility. In general, I consider that the review is in conformity with the Plos One acceptance criteria. I highlight the methodology used as a strong point of the review. Two reviewers (plus a third in case of discrepancy) are included to carry out the literature search, data extraction and risk of bias assessment. The statistical analysis of the meta-analysis is very adequate. However, some aspects should be considered.

1. In my opinion, this study is not very innovative in terms of comparing neurodynamics with static stretching. Although it is important that, compared to previous reviews, it analyses more RCTs and adds relevant assessments such as the dose-response relationship.

Authors: Thank you for the reviewer´s comment.

2. I think the criteria for the selection of studies are a bit too brief. Some more could be added, such as assessment tests to define hamstring flexibility.

Authors: Thank you for the reviewer's comment. Since the measure of effect size to be used was the standardized mean difference, a criterion for inclusion of articles could not be the use of a specific test to measure HF, but the measurement of the variable itself. However, we agree with the reviewer that the reader may be aware of some of the existing tests for measuring HF, so we have modified these lines as follows:

Study selection criteria section:

We included: (i) randomized clinical trials published in any language, (ii) comparing neurodynamic interventions with control intervention (sham, no intervention or the same protocol treatment of intervention group except neurodynamic mobilizations) or with static stretching exercises, (iii) for HF (measured using tests such as the knee extension angle (KEA) and the straight leg raise (SLR)), in (iv) healthy asymptomatic subjects with limited HF. Finally, we excluded (i) trials comparing different protocols of neurodynamic interventions and (ii) review articles, guidelines, editorials, case reports and comments.

3. I consider that the search strategy used in each of the databases should be added.

Authors: The reviewer's comment seems judicious. As suggested, we have added the search strategy used in each database in S2 table of the supplementary material.

“We searched the Cochrane, Pubmed, Scopus, WOS and Sportdiscus databases from the date of their inception to May 22nd, 2024, with no language restrictions. We used the following controlled and free-text terms: "athletic performance", performance, "sport performance", flexibility, extensibility, ROM, "nerve treatment", "nerve therapy", "neural mobilization", neurodynamic, neurodynamics, "nerve tension", "nerve stretch", "neural tension", "nerve mobilization", "neural glide", "neural treatment", "nerve gliding", nerve glide", "neural gliding", "nerve gliding exercises", “neuromobilization maneuver”, neuromobilization, "neurodynamic techniques", RCT, "randomized clinical trial", "randomized trial" and "randomized controlled trial". The search strategy used for each database is presented in S2 table of the supplementary material.”[…]

4. Inclusion criteria indicate that the sample consists of adults (line 116), although one of the studies includes subjects aged 17 (line 171). Although it is only one year difference, I think that the inclusion criteria should be corrected.

 Authors: Thank you for the reviewer's comment. We have corrected the study selection criteria section as follows:

Study selection criteria section:

We included: (i) randomized clinical trials published in any language, (ii) comparing neurodynamic interventions with control intervention (sham, no intervention or the same protocol treatment of intervention group except neurodynamic mobilizations) or with static stretching exercises, (iii) for HF (measured using tests such as the knee extension angle (KEA) and the straight leg raise (SLR)), in (iv) healthy asymptomatic subjects with limited HF. Finally, we excluded (i) trials comparing different protocols of neurodynamic interventions and (ii) review articles, guidelines, editorials, case reports and comments.

5. I think it would be appropriate to set a specific time period to define the short-term effects (lines 132-133).

Authors: The reviewer's comment seems judicious. The short-term effects of treatment are considered to be the effects of these techniques immediately after the end of the intervention programme. For this reason, it is not possible to define a specific time period (in days or months) for this variable, since each intervention programme is different in terms of duration and frequency of sessions. However, we consider this an important limitation and have taken it into account in the following section:

Limitations section:

[…]. Fifth, the included trials examined the short-term effects of neurodynamic techniques (completely after the end of the intervention regime). Therefore, future research should examine the effect of these techniques in the mid- and long-term.

In addition, we have restructured the following sentence to make it easier for readers to understand:

Our primary outcome was the efficacy (response measured by different tests) of neurodynamic interventions for improving HF. For analyses, we recorded the outcome at the end of one session (immediate effects) and at the end of the entire intervention period (short-term effects).

6. In order to keep the text uniform, ‘’hamstring flexibility`` could be changed to HF (line 213).

Authors: Thank you for the reviewer's comment. We have changed hamstring flexibility to HF.

[…]. Analyses by the type of comparison group showed significantly more HF in the neurodynamic group than in the control group (SMD = 1.35, 95% CI: 1.02 to 1.67) with a significant prI (0.89 to 1.80) and significantly more HF compared to static stretching exercises (SMD = 0.86, 95% CI: 0.13 to 1.59).

7. In statistical methodology it is described that Cohen's d is used to calculate the effect size. But the results do not express this statistic.

Authors: The reviewer's comment seems judicious. We have used Cohen's d index to calculate the standardized mean differences (SMD), so all SMD results are effect sizes of Cohen's d. We have rephrased the following sentences to clarify the above question as follows: 

We calculated the standardized mean differences (SMDs) with their 95% confidence intervals (CIs) using Cohen´s d index as the SMD method (14). Positive SMD values indicate an improvement in HF in favor of the neurodynamic intervention group. Cohen’s d values of approximately 0.2 indicate weak effects, 0.5 moderate effects, 0.8 strong effects, and above 1.0 very strong effects.

8. In my opinion, the methodology or discussion should describe what kind of neurodynamic techniques are performed in the studies included in the review; or whether the neuro

---

## [Editor Report · Decision Letter 1]

21 Jan 2025

Immediate and short-term effects of neurodynamic techniques on hamstring flexibility: A systematic review with meta-analysis.

PONE-D-24-26834R1

Dear Dr. Martínez-Vizcaíno,

We’re pleased to inform you that your manuscript has been judged scientifically suitable for publication and will be formally accepted for publication once it meets all outstanding technical requirements.

Kind regards,

César Calvo-Lobo, PhD, MSc, PT

Academic Editor

PLOS ONE

Additional Editor Comments (optional):

Authors have responded to the reviewers in a correct and satisfactory manner. Thanks for your fine work.
---

## [Editor Report · Acceptance letter]

24 Jan 2025

PONE-D-24-26834R1 

PLOS ONE

Dear Dr. Martínez-Vizcaíno, 

I'm pleased to inform you that your manuscript has been deemed suitable for publication in PLOS ONE. Congratulations! Your manuscript is now being handed over to our production team.

Kind regards, 

on behalf of

Dr. César Calvo-Lobo 

Academic Editor

PLOS ONE